# Vaginal and Uterine Microbiota of Healthy Maiden Mares during Estrus

**DOI:** 10.3390/vetsci11070323

**Published:** 2024-07-18

**Authors:** Ana Gil-Miranda, Benjamin Caddey, Daniela Orellana-Guerrero, Hanna Smith, Juan C. Samper, Diego E. Gomez

**Affiliations:** 1Department of Clinical Studies, Ontario Veterinary College, University of Guelph, Guelph, ON N1G 2W1, Canada; mirandaa@uoguelph.ca; 2Department of Production Animal Health, Faculty of Veterinary Medicine, University of Calgary, Calgary, AB T2N 1N4, Canada; benjamin.caddey@ucalgary.ca; 3Veterinary Medical Teaching Hospital, University of California, Davis, CA 95616, USA; daorellana@ucdavis.edu; 4Burleson Animal ER, Weatherford, TX 76087, USA; ridelikeagurl@gmail.com; 5Department of Large Animal Clinical Sciences, School of Veterinary Medicine and Biomedical Sciences, Texas A&M University, College Station, TX 77843, USA; jsamper@cvm.tamu.edu

**Keywords:** clitoral fossa, metritis, cervix, fertility

## Abstract

**Simple Summary:**

This study explored the different types of bacteria found in the reproductive tract of healthy maiden mares during their estrus cycle. Vaginal, uterine, clitoral fossa, and perineal skin swabs were collected from 12 mares, and the bacterial communities were characterized using a high-throughput sequencing methodology. The types and number of bacteria detected in the uterus and vagina were similar to each other but different from those in the clitoral area and skin. Specific bacteria, such as those from the families Streptococcaceae and Staphylococcaceae, were common in the uterus and vagina, but Lactobacillaceae were rare. Some bacteria associated with reproductive diseases were also found in these healthy mares, suggesting that these bacteria are naturally present even in healthy animals.

**Abstract:**

This descriptive cross-sectional study compared the microbiota of the uterus, vagina, clitoral fossa (CF), and perineal skin in healthy maiden mares during estrus. Twelve synchronized, healthy maiden mares (3–4 years old) from one single recipient mare herd were included. Microbial communities were characterized by amplifying the V3–V4 region of the 16S rRNA gene using the Illumina MiSeq platform. The uterine and vaginal microbiota had significantly lower richness (Chao-1) than the skin (*p* < 0.05). The uterine and vagina bacterial composition was similar in presence and abundance and could be differentiated from that of the CF and perineal skin. The microbial composition (Jaccard and Bray–Curtis distances) significantly differed across body-site locations (*p* < 0.05), which explained approximately 14% and 19% of the variation in microbial composition for Jaccard and Bray–Curtis distances, respectively. Firmicutes, Actinobacteria, Proteobacteria, and Bacteroidetes were the dominant taxa in the uterus and vagina, with higher proportions of Proteobacteria in the vaginal samples compared to the uterine samples. Streptococcaceae and Staphylococcaceae were present in high abundance in the uterine and vaginal samples, while Lactobacillaceae were not (<10%). We demonstrate that the uterine and vaginal microbiota of healthy maiden mares during estrus is similar but both distinct from that of the CF and perineal skin.

## 1. Introduction

Vaginal and uterine bacterial microbiota are an increasingly important area of study in veterinary medicine [1]. In humans, uterine microbiota play an essential role in fertility, reproductive homeostasis, and health [2]. In humans and cattle, specific dominant taxa in the female reproductive tract are associated with reproductive health [3,4,5]. However, in humans, the vaginal and uterine microbial core composition remains controversial due to the subjects’ inclusion criteria, differences in studies’ protocols, and variations in laboratory methodology and bioinformatic analyses. Nevertheless, the acquired knowledge has provided insights into the influence of microbiota alterations on reproductive health and the predisposition to disease [6,7]. Alterations in the normal vaginal and uterine bacteria are associated with metritis or endometritis, which are major causes of infertility in women [8], cattle [9], and equids, including jennies [10] and mares [11]. Although significant research has been conducted on the vaginal and endometrial microbiota in humans and cattle, only a few published studies have investigated the reproductive microbiota of mares [12,13,14,15,16,17,18].

In mares, the most abundant phyla within the uterus and vagina are Firmicutes, Bacteroidetes, Proteobacteria, and Actinobacteria [12,13,14]. In contrast, there is no marked dominance at the genus level. However, a recent study reported that *Lactobacillus*, *Escherichia/Shigella*, *Streptococcus*, *Blautia*, *Staphylococcus*, *Klebsiella*, *Acinetobacter*, *and Peptoanaerobacter* are detected at different levels of abundance during estrus in mares from various geographic locations [15]. There is controversy over the hormonal influence of estrogen on uterine and vaginal microbial composition during estrus [16,17,18]. In cattle, different sites of the reproductive tract have a unique microbial population, and cows, during estrus, sustain a more diverse and richer vaginal microbiota compared to uterine microbiota [16]. Both organs share similar bacterial communities, but the abundance of the communities in both vaginal and uterine samples can vary substantially [17]. In Arabian mares, vaginal microbiota remains consistent throughout the estrous cycle. However, mares in anestrus had higher microbiota diversity than during estrus, and both organs had different uterine microbial communities [18]. Despite these recent findings in equine reproductive tract microbiota, variations in microbial abundance have been reported in studies. Also, it is unknown whether the vagina, the uterus, the skin, and the clitoral fossa (CF) share the same microbiota. Therefore, the objective of this study was to describe and compare the vaginal, uterine, perineal skin, and CF microbiota of normal young maiden mares during estrus. We hypothesize that the uterine and vaginal microbiota share a common microbiota during estrus, but both sites differ from the perineal skin and CF.

## 2. Materials and Methods

### 2.1. Ethics Statement

The research protocol was reviewed and approved by the University of Florida’s Institutional Animal Care and Use Committee (IACUC # 202011054). 

### 2.2. Farm Management and Mares 

Samples were collected from a farm near Ocala, Florida, during the fall season. The experimental group consisted of 12 maiden thoroughbred mares between 3 and 4 years of age. These mares were selected from a large group of embryo-transfer recipient mares housed in 2 large pastures, fed coastal hay without grain supplementation ad libitum. All mares were dewormed routinely every three months and were up-to-date with the recommended vaccines. All mares had their vital parameters within normal limits and had good perineal conformation with no history of reproductive problems (maiden). Ultrasonographic examinations and rectal palpations of the reproductive tract were normal, without uterine discharge or intraluminal fluid (>1 cm) accumulation. None of the mares sampled had received any systemic or intrauterine antimicrobial therapy. Prior to sampling, all mares were administered 10 mL (22 mg) of altrenogest, 0.22% solution PO (Regu-mate^®^ synthetic progestin, Merck Animal Health, Rahway, NJ, USA) for ten days. Three days later, the mares received a single injection of 10 mg of dinoprost tromethamine (Lutalyse^®^ injection luteolytic agent, Zoetis Inc., Parsippany, NJ, USA) intramuscularly to induce estrus. All 12 mares were examined by transrectal palpation and ultrasonography to ensure they were in estrus with the presence of endometrial edema and a dominant follicle. Mares with follicles > 30 mm in diameter without corpus luteum in either ovary or uterine edema and a flaccid cervix on rectal palpation were sampled. 

### 2.3. Sample Collection 

The mares were led individually into stocks, and their tails were wrapped in a plastic tail bag and tied to the side. For each mare, two sterile culture swabs (BactiSwab^®^) were obtained from the perineal region without washing in between. The vulva was then washed with tap water and a dilute iodine solution. A Double-Guarded Uterine Culture Swab was introduced to the cranial vagina. To avoid vaginal contamination of the swab, a plastic sheath (first layer of protection) encasing a pipette (second layer of protection) covering the sterile swab was directed into the cervix. Inside the cervix, the pipette was pushed deep into the cervix, and once in the cervix, the sterile swab was pushed into the uterus. Once inside the uterus, the sterile cotton swab was exposed to the endometrial tissue. Before removal, the swab was pulled back inside the pipette while the pipette was still inside the cervix/uterus to avoid contamination with the vaginal mucosa. The perineal region was rewashed, and two sterile swabs were obtained from the vagina in a similar sterile fashion. Two additional sterile swabs were then taken from the CF. In addition, one air control swab was collected by exposing the swab to the environment for 30 s. In total, 12 samples per mare were obtained. Each swab was transferred to an empty sterile 15 mL conical polypropylene centrifuge tube, transported to the laboratory within 4 h, and stored at −80 °C until processing.

### 2.4. DNA Extraction and V3–V4 Sequencing 

DNA extraction and molecular profiling were carried out using established protocols, as described previously [19]. Reagent blanks and positive control samples were included in each set of DNA extractions and monitored for quality control. Amplification of the V3–V4 region of the 16S rRNA gene was completed by PCR [19] and using Illumina adapted primers, as described previously [20]. PCR products were observed on a 1.5% agarose gel. Positive samples were normalized using the SequalPrep normalization kit (ThermoFisher, Norristown, PA, USA) and sequenced on the Illumina MiSeq platform at the McMaster Genomics Facility.

### 2.5. Data Analysis 

Raw sequence data were trimmed of adapter sequences and PCR primers and quality-filtered using an average quality score cut-off of 30 and a minimum read length of 100 bp using Cutadapt v.1.18 [21]. Sequence variants were then resolved from the trimmed raw reads using DADA2 [22] for each Illumina run, error rates were learned separately for each run, and sequences were denoised to produce amplicon sequence variant (ASV) count tables. Chimeric sequences were removed, and taxonomy was assigned using the DADA2 implementation of the RDP classifier [23] against the SILVA v.138.1 database [24]. 

Sequences assigned as chloroplast or mitochondria were removed from all analyses. Samples with less than 500 sequencing reads were also removed from downstream analysis. To compare the richness, evenness, and diversity of the microbial composition between samples, ANOVA was performed on a number of observed ASVs, Chao1, Shannon’s diversity, and Fisher’s alpha index. A post hoc analysis of alpha diversity was performed using Tukey’s Honest Significant Difference. To analyze differences in microbial composition among samples, they were rarefied to 750 sequences. Jaccard and Bray–Curtis distances were calculated between samples, and differences in the microbial composition between samples were calculated using permutational MANOVA (PERMANOVA), with 999 permutations. All diversity analyses were performed using the vegan R package v. 2.5.6. Differential abundance analysis was conducted with the DESeq2 R package v. 1.26.0. Unless otherwise specified, a *p*-value of less than 0.05 was considered statistically significant for all comparisons. All analyses were conducted in R v. 3.6.3. 

## 3. Results

### 3.1. Overall Sequence Analysis

After quality filtering and DADA2 processing, 1,041,162 sequences were taxonomically classified. Mean sequences analyzed per group were vaginal, 10,479 (SD 7711); uterine, 22,272 (SD = 17,452); perineal, 23,719 (SD = 14,587); CF, 33,853 (SD = 27,969), and control air samples, 2240 (SD = 2418). The number of read counts was significantly higher in CF samples than vaginal and control air samples (*p* < 0.05). No other differences were identified. Samples with less than 500 sequence reads were discarded and resulted in one uterine, one vaginal, and three air samples being removed from the analysis. 

### 3.2. Microbial Diversity of the Equine Reproductive Tract

Alpha diversity measures show that perineal swabs had significantly higher species richness than vaginal, uterine, and CF samples (*p* < 0.05). However, the diversity and evenness of the perineal, uterine, and vaginal bacterial communities were significantly higher than the CF and air swabs (*p* < 0.05, Figure 1C). Perineal swabs also contained significantly more diverse bacterial communities than all other sampled sites, according to Fisher’s diversity index (*p* < 0.05) (Figure 1D). 

### 3.3. B-Diversity Analysis

Principal coordinate analysis on both Jaccard and Bray–Curtis distances identifies distinct sample clusters based on microbial composition (Figure 2A,B). Air, CF, and perineal samples all clustered independently and differed significantly from vaginal and uterine microbiota (*p* < 0.05), whereas uterine and vaginal samples clustered together (*p* > 0.05), regardless of the distance metric (Figure 2A,B). Body-site location differences explained approximately 14% and 19% of the variations in microbial composition for Jaccard and Bray–Curtis distances, respectively (Appendix A). 

### 3.4. Taxonomic Composition of Equine Reproductive Anatomy 

The taxonomic classification rates at the phylum, class, order, family, and genus levels were 89.7%, 89.3%, 87.9%, 79.2%, and 58.8%, respectively. At the phylum level, all sample groups predominantly consisted of Firmicutes, Actinobacteria, Proteobacteria, and Bacteroidetes (Figure 3A). The CF contained a higher relative abundance of Epsilonbacteraeota and Fusobacteria than other sampled sites (Figure 3A), while the perineal microbial communities had a relatively higher proportion of Actinobacteria (Figure 3A). The uterine and vaginal communities appeared similar at the phylum-level grouping, except for vaginal samples containing a higher abundance of Proteobacteria (Figure 3A). The CF samples had a higher relative abundance of Aerococcaceae, Campylobacteraceae, Leptotrichiaceae, and Porphyromonadaceae families than the other samples (Figure 3B). Perineal swabs had higher relative abundances of Brevibacteriaceae and Corynebacteriaceae (Figure 3B). Uterine and vaginal swabs both had relatively minor differences in microbial composition at a family-level taxonomy (Figure 3B).

Due to the low classification rate at the genus level, the differential abundance analysis was conducted at the family level. At this taxonomy level, relative differences in microbial composition become apparent between air swabs and mare swabs. Differential abundance analysis determined bacteria spanning multiple phyla as significantly more abundant in mare samples than in swabs exposed to the environment (Figure 4). Of the bacterial families that had higher relative abundance in mare vaginal, uterine, and CF samples, Aerococcaceae, Peptostreptococcaceae, Leptotrichiceae, Corynebacteriaceae, Campylobacteraceae, and Porphyromonadaceae were all significantly more abundant than in the air samples (*p* < 0.01) (Figure 4). In addition, Streptococcaceae and Staphylococcaceae were abundant in the mare’s uterine and vaginal samples (*p* < 0.01) (Figure 4).

## 4. Discussion

Here, we describe the microbiota of the uterus, vagina, CF, and perineal skin of healthy young maiden mares during estrus. The microbiota composition of the CF and perineal skin was significantly different in alpha- and beta-diversity than vaginal and uterine microbiota, but the uterine and vaginal samples clustered together based on Jaccard and Bray–Curtis distances, indicating that the bacterial composition in the vagina and uterus was similar in the presence and abundance of the different taxa and can be distinguished from the CF and perineal skin. This finding is consistent with previous studies in mares and jennies showing that endometrial and cervical microbiota are similar [10,18,25] during their fertile period. This finding is also in line with previous high-throughput sequencing studies revealing that a small proportion of shared bacterial community exists in the vagina and uterus of healthy cows [5,26]. 

In mares, the cervix exhibits a higher abundance of microbiota than the endometrium suggesting a gradual transition and reduction in microbiota from the vagina to the endometrium could occur [14]. A reasonable explanation of microbiota-sharing between the mare’s uterus and vagina is likely due to the communication between the two compartments during the estrus phase, attributed to high estrogen levels, which causes the cervix to lose tone [27] and could allow microorganisms to move between both sites. The shared taxa present in the mare’s uterus and vagina likely indicate the existence of microbial interaction between both reproductive body sites, while the distinct microbes found in each of these organs reflect the unique physiological conditions, as demonstrated in cattle [28].

The taxonomic composition of the vaginal and uterine microbiota of the mares included in our study at the phylum level was similar to that previously reported in mares which include Firmicutes, Actinobacteria, Proteobacteria, and Bacteroidetes [12,13,14]. However, variations in the dominant phylum and genera exist among studies, with some showing dominance during estrus of Firmicutes (32.03%), Bacteroidetes (31.98%), Actinobacteria (8.15%), and Proteobacteria (3.91%) in the vagina of healthy mares [12], and others showing Proteobacteria (69.9%), Firmicutes (21.1%), and Bacteroidetes (7.8%) as the most predominant phylum present in the endometrium of healthy mares [29], with distinct variations at the genus level [12,14,15,29]. Reasonable explanations for this difference might lay in factors such as mare selection, stress, diet, management practices, recent medical conditions before sample collection, and geographical location, which can influence the microbiota of different body sites [3,15,30,31,32]. Although most studies use high-throughput sequencing technology analysis, the methodology and materials for DNA extraction, PCR amplification and sequencing, and bioinformatic analyses vary substantially among studies, which can also explain some variability between the results [33,34,35,36].

Taxa belonging to the Firmicutes phylum, including Ruminococcaceae, Planococcaceae, Peptostreptococcaceae, Lactobacillaceae, Lachnospiraceae, Clostridaceae, and Aerococaceae, were the main taxa identified in the vaginal, uterine, and CF samples in our study, which is consistent with previous research conducted on the reproductive tract microbiota of women, primates, cattle, ewes, guinea pigs, and wild baboons [5,14,37,38,39]. In addition, Epsilonbacteraeota was a dominant phylum in the CF samples. Ruminococcaceae, Planococcaceae, Peptostreptococcaceae, Lactobacillaceae, Lachnospiraceae, Clostridaceae, Aerococaceae, and taxa belonging to Epsilonbacteraeota phylum are also dominant taxa in the fecal microbiota of healthy horses [40,41,42] and grain-fed cattle [43]. This suggests that in equids and other mammals, the bacterial colonization of the vagina might be associated with fecal contamination [44,45]. Perhaps due to their anatomical conformation, bacteria could ascend from the vagina to the uterus in mares, as observed in humans and other species [46,47]. However, the impact of fecal microbiota on the colonization and establishment of the vaginal and uterine microbiota of the mares needs to be further investigated.

Consistent with previous studies, the Epsilonbacteraeota phylum was abundant in the CF samples [12]. Proteobacteria was a phylum with a high abundance in the mare’s uterus and vagina, with vaginal samples containing a higher proportion of taxa belonging to this phylum. Proteobacteria are reported to be abundant in the vaginal microbiota of healthy mares and bitches [48,49]. Proteobacteria may play a role in maintaining vaginal health and balance, promoting a healthy reproductive tract, as in different species [2].

In our study, Streptococcaceae and Staphylococcaceae were found to be abundant in the vaginal and uterine samples. In healthy mares, *S. zooepidemicus* and *Escherichia coli* can be isolated from the CF and vestibule [50]. Contrarily, *Escherichia* or *Klebsiella* were not identified in high abundance, neither in the vagina nor in the uterus in the mares included in this study. *Escherichia* and *Klebsiella* have been identified in various abundances in the uterus of healthy mares [14,15,29] and mares with endometritis [51]. Although taxa belonging to the family Streptococcaceae and *E. coli* reside in the reproductive tract of healthy female equids [50], they are known to be potential reproductive pathogens, provoking endometritis [11,51]. These findings suggest that some bacteria typically associated with disease are normal inhabitants of the reproductive tract of mares. However, under certain conditions, such as inflammation associated with insemination, specific reproductive strains could colonize the uterus and cause bacterial endometritis [51,52,53,54,55].

The uterine and vaginal microbiota had significantly lower richness than the perineal skin and CF samples, suggesting that the reproductive tract of healthy mares during estrus has fewer taxa than the other external body parts. In humans, reduced vaginal richness and diversity are linked to positive reproductive health and pregnancy outcomes [56,57]. However, increased vaginal bacterial richness can lead to inflammation and alterations in the local immune system disrupting vaginal homeostasis and predisposing to disease [58]. Thus, having a low bacterial richness and diversity in the reproductive tract in mares could be favorable, as it could be associated with a lower risk of infection and potentially higher pregnancy rates. However, this hypothesis needs to be further investigated.

*Lactobacillus* is present in the vaginas of various species, being the dominant genus in the vaginal microbiota of women [59]. Lactic acid bacteria have a fundamental role in human vaginal microbiota, and their presence is beneficial for a healthy and balanced reproductive state in women [60]. *Lactobacillus* aids in maintaining vaginal health in women by limiting the colonization of the reproductive tract by opportunistic pathogens [61,62], avoiding the overgrowth of other bacterial species, and thus limiting secondary infections [63]. However, in agreement with previous studies [12,15,50,64,65], we showed a low abundance of *Lactobacillus* in the vagina of the mares. The variations in the abundance of *Lactobacillus* can be explained by the differences in the acidity levels of the vagina across various species. Compared to the vaginal pH in mares, women generally have a lower vaginal pH, ranging from 3.8 to 5.0 [35]. This indicates that factors such as pH contribute to the microbial homeostasis of the reproductive tract [12]. Also, the lack of a dominant *Lactobacillus* in healthy mares suggests that other microorganisms might have a more significant role in maintaining the reproductive health of equids. The marked difference between humans and mares indicates that specific microbiota are present in the reproductive tract of each species, and researchers should be cautious when extrapolating the results of studies from one species to another.

It is worth noting that the selected group of young maiden mares was chosen to minimize variability in the results and has never been investigated before. Additionally, the rationality of sampling this group of mares during estrus was based on the uterine immune response [66]. During estrogen dominance (estrus), the uterine immune response is more effective, contrary to when the uterus is under the influence of progesterone (diestrus), increasing the risk of infection [52,66] and potentially modifying the microbiota present during sampling. Although most high-throughput studies use healthy mares, selecting mares that have never been bred from or treated (maiden) is relevant to defining normal and abnormal reproductive microbiota in horses.

This study has several limitations. First, only mares from the same farm, subject to similar diets and management practices, were included, which could have influenced the presence of certain microbiota at the time of sampling. Also, this challenges the extrapolation of our results to a different population of mares. Additionally, the assessment of microbiota samples was performed during estrus only. Therefore, it is uncertain whether our results could be generalized to a different phase of the estrous cycle. Despite the limited sample size in this study, it was enough to identify significant differences between samples. In the future, a larger number of mares sampled during estrus and diestrus should be included to identify whether hormonal shifts influence the microbiota present in the reproductive tract of healthy mares. Additionally, due to the low biomass present in the reproductive tract, it is often challenging to differentiate between microbes present in low quantities from those that result from contamination when sequencing, in addition to other limitations of this technique. In the future, shot-gun studies should be directed toward providing better resolution beyond the family level and, ideally, sufficient knowledge of the functional microbiota in the reproductive tract of mares.

## 5. Conclusions

This study characterized endometrial, vaginal, CF, and perineal skin microbiota in healthy maiden mares during estrus using high-throughput sequencing. Like previous studies in women and cattle, the bacterial communities differed from the previously described taxa noted from culture-based studies. Also, the uterus and vagina were found to have similar microbial communities, but they were different than the CF and perineal skin regions. We also show that the microbial composition of the reproductive tract of mares differs from that of other mammalian species, particularly women and cows. The findings of this study support the hypothesis that the microbial composition of the uterus in mares is not a completely sterile environment. Furthermore, it raises the question about microbiota imbalances being the cause of endometritis.

## Figures and Tables

**Figure 1 vetsci-11-00323-f001:**
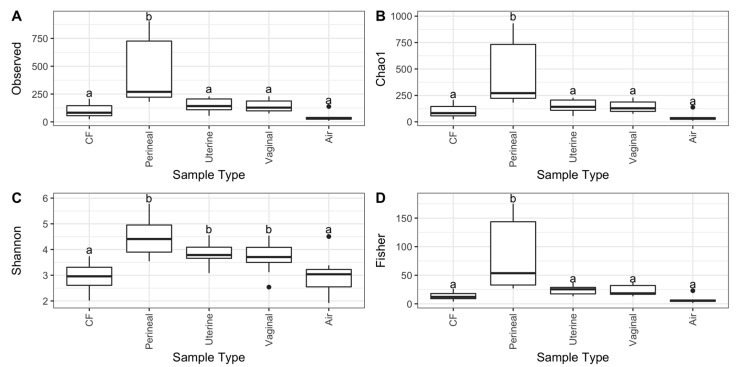
Alpha diversity metrics of uterine, vaginal, clitoral fossa (CF), and perineal skin samples from healthy mares during estrus. Indices measured were (**A**) observed ASV count, (**B**) Chao1 (richness), (**C**) Shannon’s index, and (**D**) Fisher’s alpha. Different letters for the same alpha diversity measure represent a significant difference (*p* < 0.05).

**Figure 2 vetsci-11-00323-f002:**
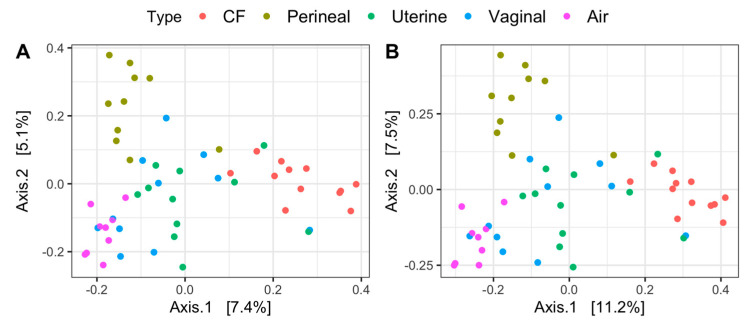
Principal coordinates analysis distances ((**A**) Jaccard and (**B**) Bray–Curtis dissimilarities) of the uterine, vaginal, clitoral fossa (CF), and perineal skin samples from healthy mares during estrus. Samples are colored according to site location.

**Figure 3 vetsci-11-00323-f003:**
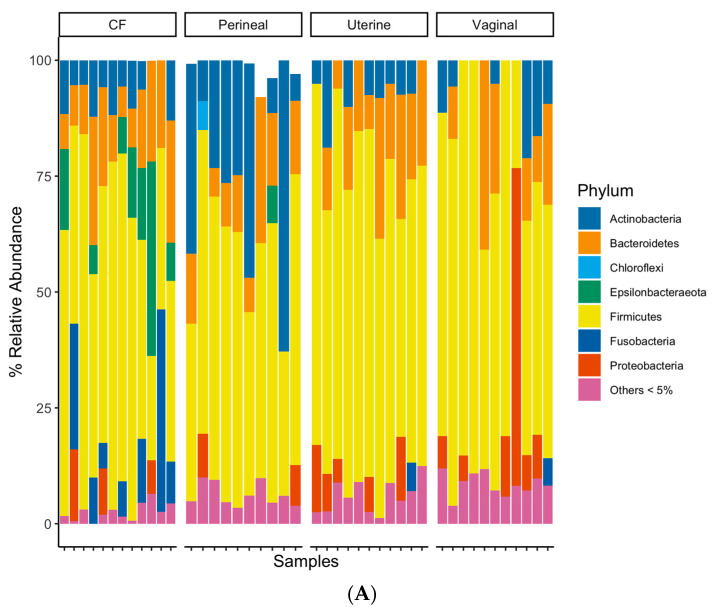
(**A**) Bacterial relative abundance for each mare and swab location. Relative abundance was calculated based on taxonomically classifiable bacteria at each taxonomic rank analyzed. Bacteria were grouped at the phylum level, and bacteria under 5% relative abundance for each swab location were grouped for clearer visualization. (**B**) Bacterial relative abundance for each mare and swab location. Relative abundance was calculated based on taxonomically classifiable bacteria at each taxonomic rank analyzed. Bacteria were grouped at the family level, and bacteria under 10% relative abundance for each swab location were grouped for clearer visualization.

**Figure 4 vetsci-11-00323-f004:**
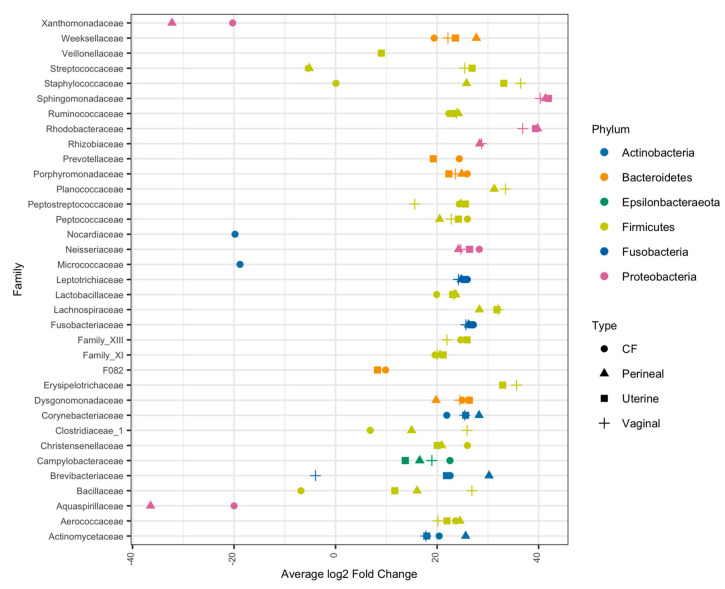
Differential abundance analysis of bacterial families for each mare swab location against air swab abundance. Differential abundances were calculated using DESeq2, and average log2 fold change is compared to air swab abundance. Only bacteria that were significantly more/less abundant in mare swab locations (*p* < 0.01) compared to air swabs are shown. Bacterial families were colored based on phyla taxonomy, and dot shape represents the mare swab location.

## Data Availability

Publicly available datasets were analyzed in this study. This data can be found in the NCBI database, bioproject number PRJNA1121906.

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
