# Peer review of "Vaginal and Uterine Microbiota of Healthy Maiden Mares during Estrus"

_vetsci, 2024, doi:10.3390/vetsci11070323_

Round 1

Reviewer 1 Report

Comments and Suggestions for Authors

Presented here was an enticing paper comparing the vaginal, uterine, clitoral fossa, and surrounding skin in healthy maiden mares. It was found that the uterine and vaginal are similar to each other but different from clitoral fossa and surrounding skin. This paper has merit and will engage interest to certain audience. I accept this publication with minor revision. Please see below.

Line 31 change embryo transfer to recipient mare herd for clarification

Lines 33-35 add by what statistical method that they are different to this sentence

Line 61 add citations

Line 163-164 Were their significant differences in the read counts found between the body sites?

Line 186 Remove sentence about visually clustered only present statistically significant differences

Lines 204-214 Include Mean, SD, and p-values for differences between phyla

Figure 3A One of the vaginal samples appears different than the rest please comment on if it was due to contamination or an outlier in this study

Results include paragraph about similarities and differences at the genus level especially to make connection between diseased bacteria within a healthy uterus which was previously stated

Line 258-261 needs clarification because line 189 says microbial composition was significantly different across body sites but here it states vagina and uterus was similar. Please clarify by which statistical test they were similar vs different

Line 297 Firmicutes is a very broad phyla, would need genus level information to make the statement in this study

Line 320 and 369 are similar consider combining paragraphs

Line 392-394 Define low biomass here because sequence reads of uterus was higher than vagina in this study or make statement broader such as due to the reproductive tract’s low biomass

Author Response

The reviewer’s comments were immensely helpful, and we appreciate such constructive feedback regarding our original submission. After addressing the issues raised, we feel the quality of the paper is much improved. Please find below our response to the reviewer’s comments.

Reviewer 1.

Presented here was an enticing paper comparing the vaginal, uterine, clitoral fossa, and surrounding skin in healthy maiden mares. It was found that the uterine and vaginal are similar to each other but different from clitoral fossa and surrounding skin. This paper has merit and will engage interest to certain audience. I accept this publication with minor revision. Please see below.

Line 31 change embryo transfer to recipient mare herd for clarification

Response: Changed as suggested. Line 28

Lines 33-35 add by what statistical method that they are different to this sentence

Response: Added as suggested. Line 33

Line 61 add citations

Response: Added as suggested. Line 57.

Line 163-164 Were their significant differences in the read counts found between the body sites?

Response: The number of read counts was significantly higher in CF samples than vaginal and control air samples. A sentence was added to the manuscript. Line 159 – 160.

read counts:  Tukey multiple comparisons of means 95% family-wise confidence level

Fit: aov(formula = lm(Count ~ Type, data = df))$Type

                       diff       lwr        upr     p adj

Perineal-CF      -10134.183 -30437.79  10169.419 0.6214573

Uterine-CF       -11580.947 -31374.75   8212.860 0.4688493

Vaginal-CF       -22436.492 -42230.30  -2642.685 0.0190120

Air-CF           -30991.361 -51901.17 -10081.548 0.0010445

Uterine-Perineal  -1446.764 -22165.62  19272.089 0.9996451

Vaginal-Perineal -12302.309 -33021.16   8416.543 0.4539139

Air-Perineal     -20857.178 -42644.70    930.349 0.0666714

Vaginal-Uterine  -10855.545 -31075.07   9363.984 0.5538652

Air-Uterine      -19410.414 -40723.67   1902.841 0.0900662

Air-Vaginal       -8554.869 -29868.12  12758.387 0.7858587

Line 186 Remove sentence about visually clustered only present statistically significant differences

Response: changes as suggested. Line 183 – 184 and 185- 187.

Lines 204-214 Include Mean, SD, and p-values for differences between phyla

Response: As mentioned in the statistical analysis (M&M section), comparing relative abundance was completed using differential abundance analysis at the family level. Figure 4 shows the differential abundances calculated using DESeq2, and average log2 fold change compared to air swab abundance. Only bacteria that were significantly more/less abundant in mare swab locations (P < 0.01) compared to air swabs are shown. Bacterial families and their respective phyla are presented.

Figure 3A One of the vaginal samples appears different than the rest please comment on if it was due to contamination or an outlier in this study

Response: Good question. Difficult to prove whether this sample’s microbial composition is a normal variation of the vaginal microbiota or a contaminant. However, the number of sequences obtained was higher than that on air samples, and the microbial composition is similar to other vaginal samples, although the proportion of sequences s belonging to Proteobacteria and Firmicutes appear to be different.

Results include paragraph about similarities and differences at the genus level especially to make connection between diseased bacteria within a healthy uterus which was previously stated.

Response: The authors reported the microbial composition at the family level due to the lower classification rate at the genus level (see table below). This is a well-known limitation of 16S RNA gene sequencing methodologies. A sentence was added to the results to make the reader aware of the reasons for presenting the data at the family level. A sentence was added to the results and discussion of the manuscript.

Table. Percent of reads classified at each taxonomic rank

Phylum

Class

Order

Family

Genus

89.7%

89.3%

87.9%

79.2%

58.8%

Line 258-261 needs clarification because line 189 says microbial composition was significantly different across body sites but here it states vagina and uterus was similar. Please clarify by which statistical test they were similar vs different

Response: Changes as suggested (see PERMANOVA results below) . Line 256 - 259

Pairwise differences for Bray and Jaccard distance (beta diversity) - PERMANOVA - only uterine and vaginal the same

Body sites

Bray-Curtis

P value

Jaccard 

P value

CF – Perineal

<0.001

<0.001

CF – Uterine

<0.001

<0.001

CF – Vaginal

<0.001

<0.001

CF – Air

<0.001

<0.001

Perineal – Uterine

<0.001

<0.001

Perineal – Vaginal

<0.001

<0.001

Perineal – Air

<0.001

<0.001

Uterine – Vaginal

0.996

0.992

Uterine – Air

<0.001

0.002

Vaginal – Air

0.004

<0.001

Line 297 Firmicutes is a very broad phyla, would need genus level information to make the statement in this study

Response: The sentence was modified to describe which family from the Firmicutes phylum was identified in higher abundance. The sentences have been modified to clarify our statement. Line 292 – 294 and 297 – 298. Also, a sentence indicating that the impact of fecal microbiota on the colonization and establishment of the vaginal and uterine microbiota of the mares needs to be further investigated was added to the discussion.

Line 320 and 369 are similar consider combining paragraphs

Results: changes as suggested.

Line 392-394 Define low biomass here because sequence reads of uterus was higher than vagina in this study or make statement broader such as due to the reproductive tract’s low biomass

Results: changes as suggested. We made the statement broader to highlight the reproductive tract’s low biomass. Line 370 – 371.

Reviewer 2 Report

Comments and Suggestions for Authors

The study by Gil-Miranda et al. proposed to describe the microbiota of the vagina, uterus, clitoral fossa and perineal skin from mares during estrus. The manuscript is generally well-written, but lacks a bit novelty, as studies performed in similar circumstances have been published already and are cited by the authors. This aspect needs to be better clarified both in the introduction, but also in the discussion. Nearly every finding of this manuscript is presented in the discussion together with the words "as shown before".

Other comments:

Abstract: authors clearly did not keep within the 200 words limit which is required by the journal and therefore the abstract was truncated. I recommend that authors read again instructions to authors and apply these in their submission.

Introduction: the hypothesis is clearly results driven, as it reflects exactly the conclusion of the study.

M&M: what was the "morning of sample collection"? which day after PGF2a treatment? It sounds like all mares were samples on the same day after synchronization treatment.

Results: section 3.1 should be included in the M&M under 2.2 where it is much more relevant than in the results.

I question the need of table 1, whose results can easily be included in the text of the manuscript.

Discussion: quite long and sometimes redundant. Lines 340-351 are really off topic and should be removed.

Author Response

The reviewer’s comments were immensely helpful, and we appreciate such constructive feedback regarding our original submission. After addressing the issues raised, we feel the quality of the paper is much improved. Please find below our response to the reviewer’s comments.

Reviewer 2

The study by Gil-Miranda et al. proposed to describe the microbiota of the vagina, uterus, clitoral fossa and perineal skin from mares during estrus. The manuscript is generally well-written, but lacks a bit novelty, as studies performed in similar circumstances have been published already and are cited by the authors. This aspect needs to be better clarified both in the introduction, but also in the discussion. Nearly every finding of this manuscript is presented in the discussion together with the words "as shown before". 

Other comments:

Abstract: authors clearly did not keep within the 200 words limit which is required by the journal and therefore the abstract was truncated. I recommend that authors read again instructions to authors and apply these in their submission.

Response: Thanks. The abstract was modified to keep it within the 200-word limit. Line 26 - 40

Introduction: the hypothesis is clearly results driven, as it reflects exactly the conclusion of the study.

Response: Thanks for your comment.

M&M: what was the "morning of sample collection"? which day after PGF2a treatment? It sounds like all mares were samples on the same day after synchronization treatment.

Response: the sentence was modified for clarification. Line 96 - 100

Results: section 3.1 should be included in the M&M under 2.2 where it is much more relevant than in the results.

Response: Modified as suggested. Line 88 – 91.

I question the need of table 1, whose results can easily be included in the text of the manuscript.

Response: Table 1 was moved to supplementary material (Supplementary Table 1)

Discussion: quite long and sometimes redundant. Lines 340-351 are really off topic and should be removed.

Response:  The discussion was modified and shortened, and, specifically, the paragraph recommended by the reviewer to be removed was deleted as suggested. References 58 – 63  - 71 & 72 were deleted.  

Reviewer 3 Report

Comments and Suggestions for Authors

Dear Authors,

This is an interesting, well written paper, in a very important topic in equine reproduction. The study has been carefully conducted and provides new and useful information on the uterine and vaginal microbiota of healthy maiden mares during estrus, despite being conducted on a restricted population. However, I have identified some minor issues that may improve the quality of the information presented here.

Introduction

Line 61: Please add the references also here.  

Line 62: Please delete “in mare,..”

Line 68 and line 77: Please add the references.

Material and methods

Please move the 3.1 paragraph of results (lines153-158) to the Material and methods at line 99, as inclusion criteria.

Line 115: Please specify whether the conical tube was empty or contained a medium, and include information about the transport temperature.

Please in Figure 4 add in the legend “Phylum colour”.  

Results

Lines 163-164: Please add also the results for perineal swab.

Line 171: Please change in “(P>0.05: Figure 1)”.

Discussion

Lines 300-302: Please add an explanation for the high abundance of Epsilonbacteraeota in CF.

Lines 308-309: Please add “in this study” if it is correct.

Line 310: Please replace “can be” with “have been”.

Lines 308-315: Did Staphylococcaceae ever get isolated? Please add this information.

This very recent study can help improve the discussion: Virendra, A., Gulavane, S. U., Ahmed, Z. A., Reddy, R., Chaudhari, R. J., Gaikwad, S. M., ... & Khan, F. A. (2024). Metagenomic analysis unravels novel taxonomic differences in the uterine microbiome between healthy mares and mares with endometritis. Veterinary Medicine and Science, 10(2), e1369.

Conclusion

Lines 404-406: Please replace with “The findings of this study support the hypothesis that the microbial composition of …”

Author Response

The reviewer’s comments were immensely helpful, and we appreciate such constructive feedback regarding our original submission. After addressing the issues raised, we feel the quality of the paper is much improved. Please find below our response to the reviewer’s comments.

This is an interesting, well written paper, in a very important topic in equine reproduction. The study has been carefully conducted and provides new and useful information on the uterine and vaginal microbiota of healthy maiden mares during estrus, despite being conducted on a restricted population. However, I have identified some minor issues that may improve the quality of the information presented here.

Introduction

Line 61: Please add the references also here.  

Response: added as suggested. Line 57

Line 62: Please delete “in mare,..”

Response: The authors believe the word mares is needed.

Line 68 and line 77: Please add the references.

Response: Line 68 – references were added as suggested. Line 64. Line 77 – this is a general statement that introduce our hypothesis.

Material and methods

Please move the 3.1 paragraph of results (lines153-158) to the Material and methods at line 99, as inclusion criteria.

Response: changed as suggested.

Line 115: Please specify whether the conical tube was empty or contained a medium, and include information about the transport temperature.

Response: The tube was empty.  Added as suggested. Line 117 – 18.

Please in Figure 4 add in the legend “Phylum colour”.  

Response: added as suggested.

Results

Lines 163-164: Please add also the results for perineal swab.

Line 171: Please change in “(P>0.05: Figure 1)”.

Response: changed as suggested. Line 169

Discussion

Lines 300-302: Please add an explanation for the high abundance of Epsilonbacteraeota in CF.

Response: Added as suggested. Line 297 - 300

Lines 308-309: Please add “in this study” if it is correct.

Response: changed as suggested. Line 313

Line 310: Please replace “can be” with “have been”.

Response: changed as suggested. Line 315

Lines 308-315: Did Staphylococcaceae ever get isolated? Please add this information.

Response: the sentence “taxa belonging to the family Streptococcaceae” was added to indicate that bacteria such as S. zooepidemicus have been isolated from uterine samples. Line 314 and 319

This very recent study can help improve the discussion: Virendra, A., Gulavane, S. U., Ahmed, Z. A., Reddy, R., Chaudhari, R. J., Gaikwad, S. M., ... & Khan, F. A. (2024). Metagenomic analysis unravels novel taxonomic differences in the uterine microbiome between healthy mares and mares with endometritis. Veterinary Medicine and Science, 10(2), e1369.

Response: thanks for the suggestion. The article was added to the manuscript. Reference 51. Line 533.

Conclusion

Lines 404-406: Please replace with “The findings of this study support the hypothesis that the microbial composition of …”

Response: changed as suggested. Line 384 - 385

Round 2

Reviewer 2 Report

Comments and Suggestions for Authors

The paper has just slightly been improved and the novelty of the study is still not presented.

Author Response

Reviewer 2.

The paper has just slightly been improved, and the novelty of the study is still not presented.

Response: Thanks for your comment.

Line 71 states: Despite these recent findings in the equine reproductive tract microbiota, variations in microbial abundance have been reported in studies.” Thus, additional studies are needed to identify a possible core microbiota of the reproductive tract of mares.

line 73, we stated: "Also, it is unknown whether the vagina, the uterus, the skin, and the clitoral fossa (CF) share the same microbiota."

In Table 1, presented below, we have summarized all studies published to date that investigate the microbial communities of the reproductive tract of mares using high-throughput sequencing methodologies. As noted, no published studies have compared the microbiota of the vagina, uterus, skin, and CF within the same animal. This gap highlights the novelty of our manuscript.

We have chosen to avoid the assertion that ours is the "first" study of its kind. This is why we have refrained from using phrases such as "this is the first study investigating…" in the introduction of our manuscript.

We hope this clarifies our position and the contribution of our study to the existing literature.

Table 1. Summary of main objective of studies describing the microbiota of the reproductive tract in mares.

Autor(s) Objective/hypothesis

Inclusion criteria, (n), and study’s location, and month

Sathe et al., 2017*

Hypothesize that the uterus of healthy mares is not sterile and is colonized by complex microflora.

Healthy mares in estrus and early pregnancy.

n=20. USA.

Holyoak et al., 2018

to characterize the

uterine equine microbiome in the non-pregnant mare

Healthy mares n=29, USA

Jones, 2019**

A. Describe and compare the vaginal, uterine, and fecal microbiota of the mare and stallion semen.

B. Evaluate the impact of raw or extended semen on the uterus and vagina microbiotas following insemination.

A. Healthy mares, n=16, Healthy stallion n=1, USA

B. Healthy mares n=8, PBIE mares (Persistent breeding induced endometritis)

Barba et al., 2020

Characterize the vaginal microbiota in Arabian mares using traditional culture- dependent and metagenomics and identify changes in estrous cycle.

Healthy mares in estrus and diestrus. n=8, Spain (June-July).

Thomson et al., 2022

Characterize the uterine microbiota in mares and predict its metabolic pathways.

Healthy mares in estrus., n=21, Chile (October).

Holyoak et al., 2022

Describe the endometrial microbiome of mares in different geographical locations.

Mares with no reproductive history. n=54

North America (Oklahoma, Louisiana) & Australia

Heil et al., 2023.

Explores different sampling techniques to detect uterine microbiome in mares.

Mares in estrus without signs of endometritis on cytology and negative aerobic culture.

n=15, Louisiana State, USA.

Beckers et al., 2023

Identify the microbiome in different sites of pregnant pony mares.

Pregnant mares (96-120 days of gestation length upon necropsy).

n=5, Louisiana State, USA.

Virendra, A, 2024

Evaluating the differences in uterine microbiome between healthy mares and mares with endometritis.

30 mares classified into healthy (n = 15) and endometritis (n = 15) based on their reproductive history, intrauterine fluid accumulation, gross appearance of LVL samples, endometrial cytology and bacterial culture.

*Abstract only

**Thesis project (second project).

*** Does not identify variable regions of the DNA sequencing.
